# Magnetic Field Treatments Improves Sunflower Yield by Inducing Physiological and Biochemical Modulations in Seeds

**DOI:** 10.3390/molecules26072022

**Published:** 2021-04-01

**Authors:** Irfan Afzal, Saman Saleem, Milan Skalicky, Talha Javed, Muhammad Amir Bakhtavar, Zia ul Haq, Muhammad Kamran, Muhammad Shahid, Muhammad Sohail Saddiq, Aneela Afzal, Noshin Shafqat, Eldessoky S. Dessoky, Aayushi Gupta, Joanna Korczyk-Szabo, Marian Brestic, Ayman E. L. Sabagh

**Affiliations:** 1Department of Agronomy, University of Agriculture, Faisalabad 38040, Pakistan; samansaleem80@gmail.com (S.S.); talhajaved54321@gmail.com (T.J.); amir.bakhtavar@yahoo.com (M.A.B.); kamranagr@gmail.com (M.K.); sohail1540@gmail.com (M.S.S.); 2Department of Botany and Plant Physiology, Faculty of Agrobiology, Food and Natural Resources, Czech University of Life Sciences Prague, Kamycka 129, 165 00 Prague, Czech Republic; skalicky@af.czu.cz (M.S.); gupta@af.czu.cz (A.G.); marian.brestic@uniag.sk (M.B.); 3Department of Physics, University of Agriculture, Faisalabad 38040, Pakistan; zh_uaf@hotmail.com; 4Department of Biochemistry, University of Agriculture, Faisalabad 38040, Pakistan; mshahiduaf@yahoo.com; 5Department of Sociology, PMAS-Arid Agriculture University, Rawalpindi 46300, Pakistan; anilaafzal@gmail.com; 6Department of Agriculture, Hazara University, Mansehra 21120, Pakistan; nosheenshafqat@gmail.com; 7Department of Biology, College of Science, Taif University, P.O. Box 11099, Taif 21944, Saudi Arabia; es.dessouky@tu.edu.sa; 8Department of Sustainable Agriculture and Herbology, Slovak University of Agriculture, 949 01 Nitra, Slovakia; joanna.korczyk-szabo@uniag.sk; 9Department of Plant Physiology, Slovak University of Agriculture, Tr. A. Hlinku 2, 949 01 Nitra, Slovakia; 10Department of Agronomy, Kafrelsheikh University, Kafr Elshiekh 33516, Egypt

**Keywords:** magnetically treated water, moringa leaf extract, vigor, magnetic field, oilseed

## Abstract

Magnetic seed enhancement has been practicing as a promising tool to improve germination and seedling growth of low vigor seeds stored under suboptimal conditions, but there is still ambiguity regarding the prospects for magnetism in oilseeds. Present study elucidates the potential of magnetic seed stimulation to improve sunflower germination, growth and yield. Germination and emergence tests were performed to optimize the strength of the magnetic field to sunflower seed enhancement. The seeds were directly exposed to magnetic field strengths of 50, 100 and 150 millitesla (mT) for 5, 10 and 15 min (min) and then standard germination tests were performed. Secondly, the emergence potential of untreated seeds was compared with seed exposed to hydropriming, priming with 3% moringa leaf extract (MLE), priming with magnetically treated water (MTW) for 10 min and priming with 3% MLE solution prepared in magnetically treated water (MTW + MLE). Germination, emergence, seedling growth and seed biochemical properties were used to select the best treatment for field evaluation. The results of the study revealed that magnetic seed treatment with 100 mT for 10 min and seed priming with 3% MLE solution in magnetically treated water (MTW + MLE) significantly improved emergence, crop growth rate and sunflower yield.

## 1. Introduction

Quality seed with robust germination is always an important consideration in commercial field crop production [1]. The degradation of seed components during storage occurs through damages caused by oxidizing agents, but the extent of such reactions is defined by quality of the seeds, which is affected by fluctuations in temperature and moisture [2]. Sunflower is an important oilseed crop with 54.77 million metric tons of sunflower seeds and 1.78 million metric tons of oil production [3]. Sunflower seeds have lipids composition, which reinforce the production of reactive oxygen species (ROS) and may lead to reduced germination and viability [4] during storage. On the other hand, oil concentrations, activities of superoxide dismutase, alcohol dehydrogenase, catalase, and malate dismutase decreased in sunflower seeds during storage, which result into further decline in germination over time [5]. Unfortunately, these are the aspects over which farmers have no control and they have to rely on these deteriorated seeds for crop production [6].

To improve germination properties of seeds, different seed treatments generally called seed enhancements have been practiced including seed priming, thermal treatments, seed coating and pelleting [7]. Seed priming improves germination as it triggers a range of biochemical changes such as enzyme activation, starch hydrolysis and dormancy breaking in seed [8]. Priming sets in motion germination-related activities (e.g., respiration, endosperm weakening, and gene transcription and translation, etc.) that enable the transition of quiescent dry seeds into germinating state and lead to improved germination potential [9]. Priming treatment also known to decrease the risk of abnormal seedlings emergence of vegetable seeds stored in detrimental conditions by means of inducing repair mechanism [10]. Leaf extract of moringa (*Moringa oleifera* Lam.) tree has shown its potential as natural priming agent for maize seed enhancement [11].

Safe methods for increasing productivity including use of chemicals and substitution of some of them by appropriate physical treatments [12] such as laser irradiation, gamma irradiation and magnetic field etc. [13] become attractive seed enhancement techniques. Magnetic seed treatments have been used successfully to improve germination rate and crop establishment, boost growth, development and ultimately yield of many horticultural and agronomic crops [14]. Moreover, it also reduced the attack of pathogenic diseases [15,16,17,18,19,20] during handling.

The water treated with magnetic field alters its electronic, atomic and molecular structure such as its viscosity, boiling point, solidifying and dielectric constant [21,22]. It also changes the formation of clustering structures and increasing polarizing effect of water molecules [23,24]. Magnetic field treatment makes membrane more permeable by changes the electrical charges, ion concentration and free radicals without any alteration in chemical profile of seed [25]. This free movement of ion triggers the metabolic pathway by enhancing the physiological and biochemical response [26,27]. Furthermore, magnetic field effects were dependent on field strength, duration of treatment and frequencies modulation [25,28]. Based on ion cyclotron resonance theory, some scientist reported that magnetic field influenced the transport of un-hydrated ions through membrane channels according to the combination of the magnetic field frequency and the static intensity [29]. Many studies have reported that magnetically treated sunflower seeds showed high performance in term of plant growth e.g., increased total fresh weight and root fresh weight [30]. Similarly, sunflower seed yield and oil recovery can be improved by priming the seeds with laser and magnetic field treatments before sowing [31].

Magnetic field pretreatment increased seed germination rate, seedling growth and yield [32] of cumin. The beneficial effect of magnetic seed stimulation has been associated with various biochemical, cellular, and molecular events including enzymatic stimulation and bioenergetic excitements [33] and synthesis of proteins. Ion cyclotron resonance, radical pair mechanism [34] and formation of radicle pair by cytochromes have been reviewed as magnetoreceptors in plants [35]. Only a few studies relating to the biochemical and physiological aspects of magnetic seed stimulation on oil seeds have been reported so far and concentrated mainly on cereals and root crops. The aim of this study was to investigate the effects of different priming techniques and magnetic field treatment on germination, early seedling growth, yield as well as biochemical aspects of sunflower seeds after exposure to magnetic field and seed priming have been investigated.

## 2. Results

Magnetic seed treatment, 150 mT or 100 mT for 10 min significantly reduced the mean germination time and gave maximum value of final germination percentage, while untreated seeds took maximum time to germinate and gave minimum value of germination percentage. Maximum value of germination energy was observed in magnetic seed treatment 100 mT for 10 min whereas least values were given by control treatment (Table 1).

Magnetic seed stimulation improved seedling attributes such as root length, shoot length and root-shoot ratio. Highest root and shoot lengths and root shoot ratio were measured in seeds which were magnetically treated for 10 min at both 100 mT and 150 mT (Table 2). Seed biochemical analysis showed a significant increase in α-amylase activity, total soluble sugars and reducing sugars after magnetic seed stimulation with strength of 100 mT and 150 mT for 10 min. (Table 3).

Moreover, different priming techniques significantly affected the emergence attributes of sunflower. Lowest values of mean emergence time and highest emergence energy were recorded for seeds primed with 3% solution of moringa leaf extract and magnetically treated water (MTW + MLE priming) while control treatment took maximum time to emerge and exhibited lowest emergence energy (Figure 1). Similarly, emergence index and final emergence percentage were also improved with MTW + MLE priming treatment. Furthermore, seed priming with MTW+MLE increased root length, shoot length and root shoot ratio of sunflower seed as compared to control and other priming treatments (Figure 2).

Besides physio-morphological attributes enhancements, maximum α-amylase activity was found in seeds primed with MTW + MLE followed by priming with MTW alone. Total soluble sugars and reducing sugars were also boosted in MTW + MLE priming treatment (Figure 3).

Both physical and physiological seed enhancements improved emergence of sunflower in field. Magnetic seed stimulation at 100 mT for 10 min followed by seed priming with MTW significantly reduced mean emergence time, maximized final emergence percentage, and emerge index. However, control treatment showed minimum values for both these parameters (Figure 4).

Higher crop growth rate at each measurement was recorded for the crop raised from seed primed with magnetically treated water followed by magnetic stimulation at 100 mT for 10 min as compared to other treatments. Maximum crop growth rate was measured on 60 days after sowing and then it declined onward till maturity (Figure 4). Plant achieved maximum height and carried maximum number of achenes per head which were emerged from seed treated with magnetic stimulation at 100 mT for 10 min. Similarly, an increase in head diameter was also observed by magnetic seed stimulation at 100 mT for 10 min and MTW + MLE priming. Yield attributes including 1000- achene weight, biological yield, achene yield and harvest index were significantly higher for magnetic seed stimulation at 100 mT for 10 min over other enhancement treatments (Table 4).

## 3. Discussion

The intension of researchers to enhance the productivity of crops through noninvasive and environment friendly biophysical techniques such as magnetic field treatment is an attractive approach [36,37,38]. The present study indicated that magnetic seed treatments significantly improved germination and seedling growth of sunflower. The effects of 100 mT and 150 mT for 10 min were more pronounced in synchronizing germination as exhibited in low mean germination time, higher germination index, germination energy and final germination percentage [14,39]. Improvement of germination in sunflower seeds by magnetic seed treatment might be because of changes in seed phytohormone contents particularly reduction of abscisic acid content [37] and enhancement of antioxidant capacity [40] that stimulated germination. An improvement in germination potential and seedling vigor by the influence of magnetic field in the seeds of radish [41], cumin [32] and soybean [42] had also been reported. Improvement in root and shoot lengths of magnetically treated seeds (Table 1 and Table 2) as compared to control is due to an increased rate of cell division in the root tips and earlier start of emergence as indicated by lower values of MET [7]. Magnetic treatments are assumed to enhance seed vigor by influencing life processes that involve free radicals’ production, and by stimulating the activity of carbohydrate and proteins [43,44]. An increased α-amylase activity along with contents of total and reducing sugars of high strength magnetically treated seeds was observed in this present study (Table 3). It confirms the primary role of magnetic treatment in either stimulating protein synthesis or enhancing the activities of existing enzymes [7,20], thereby producing germination metabolites in requisite amounts.

Combination of magnetically treated water and moringa leaf extract significantly improved emergence and seedling growth in the present study (Figure 1, Figure 2 and Figure 3). Improved emergence of sunflower seeds by magnetic treated water might be due to the fact that magnetic field influenced the microscopic and macroscopic properties of water by changing the dipole-moment transition, vibration states of molecules and vibration of transition probability of electrons [45]. These changes increased the soaking degree and hydrophobicity of water to materials, depressed its surface-tension force, diminished the viscosity of water and enhanced the electric conductivity of water after magnetization and enhanced α- amylase activity through breakdown of food reserves in pea seeds [46]. Earlier start of emergence as indicated by lower values of MET is the possible reason for increase of root and shoot lengths and root shoot ratio (Figure 2).

Moringa leaf extract is a rich source of plant hormones (Zeatin), antioxidants (phenolics, ascorbate), vitamins like (A, B, C), different essential minerals including K, Ca and Fe [39]. Presence of these growth promoting substances in MLE also suggest increased seedling growth. Seed priming with MLE increased root and shoot length, root shoot ratio, seedling fresh and dry mass of maize [18]. Increased α-amylase activity, total soluble and reducing sugars in sunflower seeds primed with MTW+MLE is due to the fact that magnetically treated water remarkably improved seedling vigor by increasing starch hydrolysis [11]. Higher emergence rate with reduced mean emergence time are the main contributors, which ensures an improvement of overall seedling performance, crop growth rate and yield related traits [1,39]. In the present study seed enhancements not only speed up emergence but also improved final emergence and crop growth rate, which ultimately contributed to improved economic yield of sunflower (Figure 4).

According to the original ion cyclotron resonance (ICR) model, addition of Ca^2+^ ions reduce the oscillation and induces large scale coherent regions in water which regulate the ion fluxes [29]. Magnetically induced ICR can thus alter equilibrium of biochemical reactions. Magnetic field is a kind of energy treatment, which is absorbed by electron in different molecules and then used for accelerating the seed metabolism that trigger the biochemical and enzyme reaction to help in early seed germination [13]. The higher concentration of indole-3-acetic acid and gibberellin acid as well as higher imbibition rate was observed in germinating seeds of faba bean [33]. Improved emergence of magnetically treated seeds is the outcome of high concentration of secondary metabolites, enzymatic activity, and anti-oxidative capacity [39,41]. Increased α-amylase activity, total soluble and reducing sugars have been found in magnetically treated zinnia seeds as is reported [7]. Increase in crop growth rate, chlorophyll contents and yield of maize has been observed as a result of magnetic seed stimulation [42].

Almost all the seed enhancements have positive effects on plant growth and yield contributing attributes in sunflower. However, maximum head diameter, number of achene per head, 1000-achene weight, biological yield and achene yield were observed in plants raised from magnetically treated seeds of 100 mT for 10 min. Improved yield by treated seeds as compared to untreated seeds seems to be the result of yield contributing factors i.e., head diameter, no of grains per head and 1000-grain weight which are the outcome of energetic start by early and improved emergence of magnetically treated seeds and high crop growth rate (Table 4). Incremental effects of magnetically treated water and magnetic field have been observed on yield of maize [20].

## 4. Materials and Methods

### 4.1. Lab Study

Seeds of sunflower (*Helianthus annuus* L., genotype Armoni) were collected from Ayub Agricultural Research Institute, Faisalabad, Pakistan (31.40° N, 73.04° E). Initial seed moisture contents were reported 10% on fresh weight basis. First experiment primarily focused on magnetic field stimulation. Seeds (100 g seeds for each treatment) were exposed to different strengths of magnetic field (50, 100, 150 mT) for different time duration (5, 10 and 15 min) and effect of treatments was evaluated during germination assay. In second experiment, seeds were exposed to different priming techniques i.e., hydropriming with water, priming with 3% moringa leaf extract (MLE), priming with magnetically treated water for 10 min (MTW) and priming with solution of 3% MLE and magnetically treated water (MTW+MLE). For each type of priming, the seeds were soaked for 12 h and then re-dried under shade near to their original weight and emergence potential of each treatment was compared. The untreated seed was considered as control in both experiments.

### 4.2. Germination Assay

Germination test was performed according to the guidelines issued by the International Seed Testing Association [47]. Hundred treated seeds of four replications were positioned on Whatman No. 1 filter paper wetted with distilled water and placed in an incubator (SANYO MIR-254, Japan) at 25 °C with adequate light to help seedling growth. Visible radicle protrusion was recognized as index for germination.
(1)$$\mathrm{Mean}\;\mathrm{Germination}\;\mathrm{Time}\;\left(\mathrm{MGT}\right)=\;\frac{\Sigma \mathrm{Dn}}{\Sigma n}$$
where *n* is the number of seeds which germinated on day D, and D is the number of days counted from the beginning of germination.
(2)$$\mathrm{Germination}\;\mathrm{Index}\;\left(\mathrm{EI}\right)=\;\frac{\mathrm{No}.\;\mathrm{of}\;\mathrm{germinated}\;\mathrm{seeds}}{\mathrm{Days}\;\mathrm{of}\;\mathrm{first}\;\mathrm{count}}+\;\frac{\mathrm{No}.\;\mathrm{of}\;\mathrm{germinated}\;\mathrm{seeds}}{\mathrm{Days}\;\mathrm{of}\;\mathrm{final}\;\mathrm{count}}$$

### 4.3. Emergence Test

Treated and untreated seeds with four replications were sown in plastic trays (25 seeds in each replication) containing moist sand, and were placed in a growth chamber (SANYO, Osaka, Japan) at 25 °C with continuous fluorescent light during the course of investigation. Emergence was recorded daily on the basis of appearance of cotyledons. Seedlings were harvested after two weeks and washed with deionized water and subsequently, seedlings root-shoot lengths, fresh and dry mass (after oven drying the samples at 65 °C for three days) were recorded
(3)$$\mathrm{Mean}\;\mathrm{Emergence}\;\mathrm{Time}\;\left(\mathrm{MET}\right)=\;\frac{\Sigma \mathrm{Dn}}{\Sigma n}$$
where *n* is the number of seeds which emerged on day D, and D is the number of days counted from the beginning of emergence.
(4)$$\mathrm{Emergence}\;\mathrm{Index}\;\left(\mathrm{EI}\right)=\;\frac{\mathrm{No}.\;\mathrm{of}\;\mathrm{emerged}\;\mathrm{seeds}}{\mathrm{Days}\;\mathrm{of}\;\mathrm{first}\;\mathrm{count}}+\;\frac{\mathrm{No}.\;\mathrm{of}\;\mathrm{emerged}\;\mathrm{seeds}}{\mathrm{Days}\;\mathrm{of}\;\mathrm{final}\;\mathrm{count}}$$

### 4.4. Magnetic Field Treatments

Pre-sowing magnetic treatment was carried out in the Department of Physics, University of Agriculture Faisalabad, Pakistan (31.43° N, 73.07° E) by using a magnetic seed stimulator. A glass petri dish with sunflower seeds was placed between center of the poles of electromagnets (the central part of electromagnet has a dimension 10 cm × 10 cm) [14]. In full wave rectification four diodes in a bridge configuration were used with step down transformer, which convert any AC source into pulsating DC. Four rectangular coils which are not Helmholtz coils were constructed using enameled copper wire of thickness 0.42 mm, each coil having 4000 turns. These coils are wound on two soft iron bars (electromagnet, having dimension 40 cm × 3.5 cm) kept one above the other (two coils wound on each bar), their ends held by metallic supports. These coils were connected in series in order to maintain the current same in all coils and fed through a power source using a variable transformer (0 V–260 V AC). A 50 Hz full wave rectified sinusoidal voltage was fed to the coils. When electric current is passed through these coils, a non-uniform magnetic field is produced between the air gaps of two iron bars, the magnetic field (MF) is varied by changing the current through the coils [15]. This MF was measured with the help of magnetic fluxmeter probe (ELWE 8533996, Cremlingen, Germany). The strength of MF was controlled by regulating the current in electromagnet coils (Figure 5).

### 4.5. Priming with Magnetically Treated Water

For magnetically treated water, a magnetic field was developed by two sets of thirteen magnets which were connected on each side of wooden bars and a PVC pipe (length of the pipe is 87 cm and 12.7 mm internal diameter) was passed through the magnetic field of strength 211 mT. The intensity of MF was measured along the longitudinal. When water was passed through this magnetic field, as water contain sodium chloride, it converted into ionic layers of sodium on the negative side of the magnet and chlorine on the positive side of the magnets also molecular cluster of H_2_O becomes smaller as the water passed through these magnetic field. Magnetically treated water was prepared by passing distilled water through the magnetic field for 10 min and then used to soak the seeds for 12 h. Seeds were dried at room temperature until they attained their original weight.

### 4.6. MLE Priming

For priming with MLE, fresh moringa leaves were collected and juice was extracted by a locally fabricated juice extraction machine. Solution of 3% moringa leaf extract and magnetically treated water and distilled water separately was prepared, and seeds were soaked overnight in both solutions with continuous aeration. For hydropriming, 100 g seeds were weighed by using weighing balance (Uni Block AUX220, Shimadzu Corporation, Kyoto, Japan). The weighed seeds were soaked in aerated distilled water for 12 h.

### 4.7. Germination Attributes

Germination energy (GE) mean germination time (MGT), germination index (GI) and final germination percentage (FGP) of each treatment were calculated [19]. Mean emergence time (MET) was calculated according to equation of [48], while emergence index was calculated as described by the Association of Official Seed Analysts [49].

### 4.8. Biochemical Analysis

α-amylase activity was determined in sunflower seed sample (0.1 g), extracted by using potassium phosphate buffer (pH 7.0). After preparation of sample, α-amylase activity was determined by following the modified DNS method [50]. Total soluble sugars were quantified after grinding the sample with the help of mortal pistol followed by hydrolysis with 2.5 N HCl solutions and then neutralized by sodium carbonate and made-up final volume of 10 mL by distilled water. This solution was centrifuged at 10,000× *g* and the supernatant was used for measurement of total sugars following the phenol-sulphuric acid method [51]. Total reducing sugars were measured by DNS method from the sunflower seed sample (0.1 g) extracted in 80% ethanol twice using 5 mL volume each time.

### 4.9. Field Study

Optimized magnetic field seed treatments were taken from lab study and were further evaluated in field at Crop Physiology Research Area, University of Agriculture Faisalabad, Pakistan (31.43° N, 73.07° E) on 1 February 2014. The experiment was laid out in Randomized Complete Block Design (RCBD) with three replications. Pretreated seeds were sown on ridges 75 cm apart keeping plant to plant 25 cm distance in a plot having net size of 5 m × 3 m.

Treatments include magnetically treated seed (100 mT and 150 mT for 10 min), hydropriming, priming with 3% MLE, priming with magnetically treated water for 10 min and priming with 3% MLE solution prepared from magnetically treated water (MTW) along with control. Fertilizer was applied^@^ 150:100:62 kg ha^−1^ NPK and rest agronomic practices were similar for all treatments. The average daily temperature field was 14 °C in the month of February at sowing time.

### 4.10. Growth, Morphological, and Yield Traits

For the determination of crop growth rate plants from 1 m^2^ area were harvested from each experimental unit on 15, 30, 45, 60 and 75 days after sowing. Plant height, head diameter, number of achenes per head and 1000-achene weight were determined from randomly selected three plants in each replicate and then average was taken. For determination of economic and biological yields and harvest index a five-meter row of plants was harvested at maturity. Crop was sundried for ten days before recording achene yield.

### 4.11. Statistical Analysis

Data recorded were pooled for statistical analysis to determine the significance of variance (*p* ≤ 0.05) using Statistix 8.1 version computer software (Statistix 8.1, Tallahassee, FL, USA). Two way bi-factorial ANOVA was employed for magnetically field treatments at different time intervals (magnetic field treatment × time of exposure) in first part of lab screening. For second priming experiment of lab screening, one-way ANOVA was executed to compare the treatments means following Tukey’s HSD test. While the second phase (field study) was subjected to one-way ANOVA. After the application of ANOVA, Tukey’s test was applied to ascertain the nature of the differences among the treatment. For comparison of treatment means, standard errors were computed using Microsoft Excel (https://www.microsoft.com/en-gb/microsoft-365/excel/ (accessed on 7 February 2019)). Moreover, to check the normal distribution of data, we conducted Leven’s test of homogeneity of variance using SPSS. The “significance values” obtained were higher than 0.05, indicating non-significance of the model. Which means, the variances were homogenous, depicting normal distribution of data.

## 5. Conclusions

Magnetic stimulation of sunflower seeds in a magnetic field of 100 mT for 10 min proved most effective in improving the emergence, seed biochemical attributes, agronomic, growth and yield traits. Seed priming with 3% MLE solution in magnetically treated water (MTW + MLE) also significantly improved emergence, seed biochemical activities, crop growth rate, and yield of sunflower thus making magnetic seed stimulation more practical and convenient to use.

## Figures and Tables

**Figure 1 molecules-26-02022-f001:**
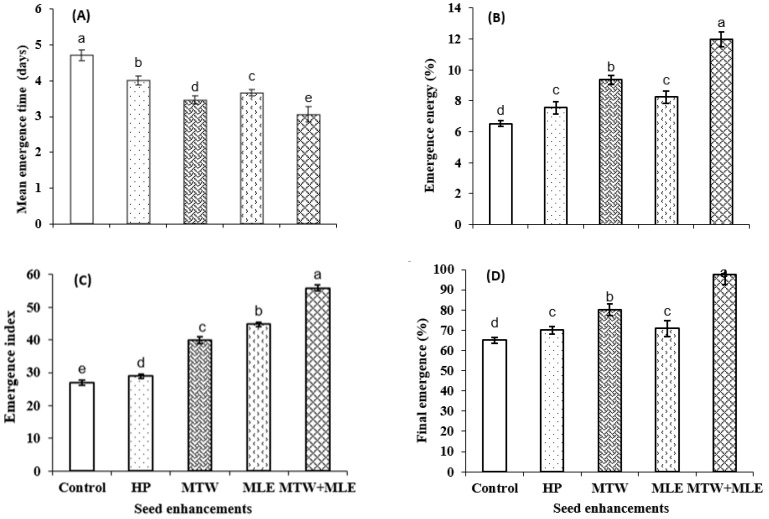
Effect of magnetic seed stimulation on emergence of sunflower Armoni genotype during emergence test. (**A**) Mean emergence time; (**B**) Emergence energy; (**C**) Emergence index; (**D**) Final emergence (%). HP; Hydropriming, MTW; magnetically treated water, MLE; moringa leaf extract. Lettering (a–e) on the bars shows mean separation within columns by one-way Tukey’s HSD at *p* ≤ 0.05.

**Figure 2 molecules-26-02022-f002:**
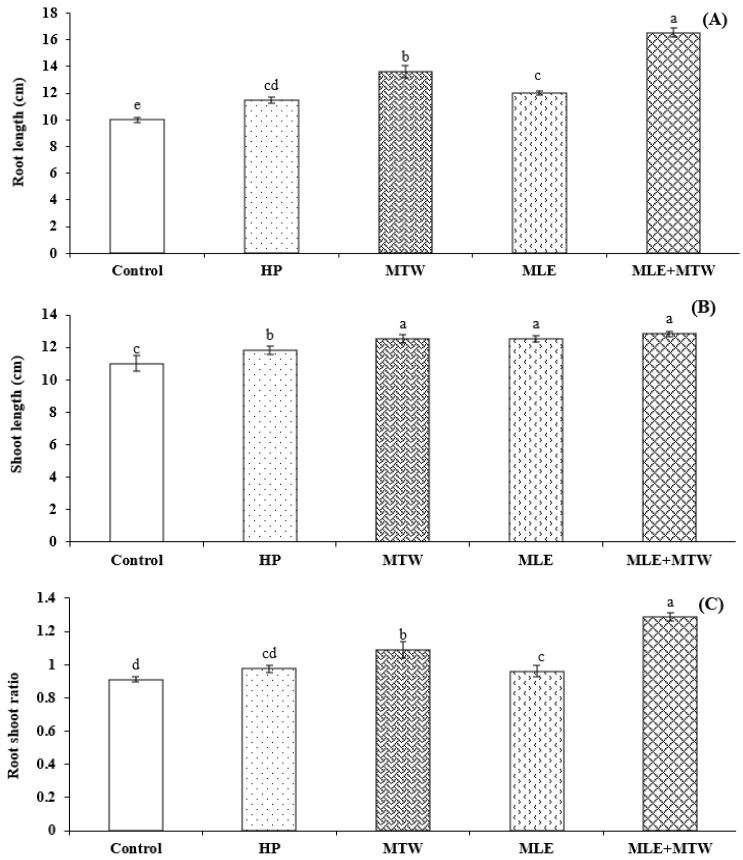
Effect of magnetic seed stimulation on seedling growth of sunflower Armoni genotype during emergence test. (**A**) Root length; (**B**) Shoot length; (**C**) Root shoot ratio. HP; Hydropriming, MTW; magnetically treated water, MLE; moringa leaf extract. Lettering (a–e) on the bars shows mean separation within columns by one-way Tukey’s HSD at *p* ≤ 0.05.

**Figure 3 molecules-26-02022-f003:**
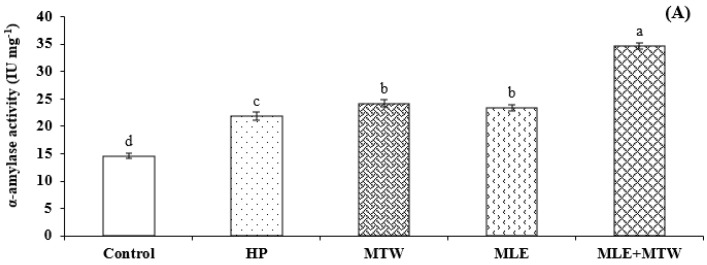
Effect of magnetic seed stimulation on biochemical attributes of sunflower Armoni genotype during emergence test. (**A**) α-amylase specificity; (**B**) Total soluble sugars; (**C**) Reducing sugars. HP; Hydropriming, MTW; magnetically treated water, MLE; moringa leaf extract. Lettering (a–e) on bars shows mean separation within columns by one-way Tukey’s HSD at *p* ≤ 0.05.

**Figure 4 molecules-26-02022-f004:**
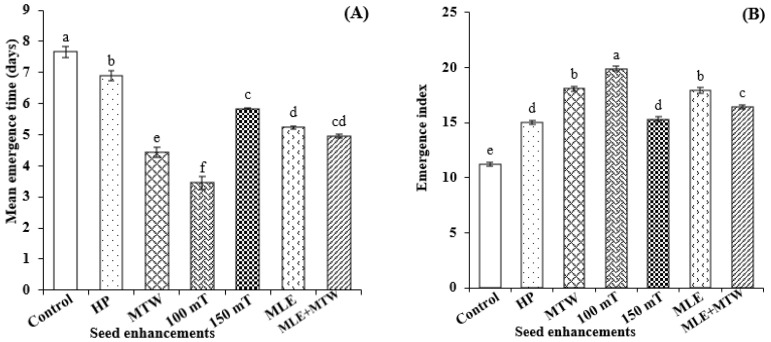
Effect of seed enhancements on emergence and crop growth rate of sunflower. (**A**) Mean emergence time; (**B**) Emergence index; (**C**) Final emergence (%); (**D**) Crop growth rate (CGR1 (30 days after swing); CGR2 (45 days after sowing); CGR3 (60 days after sowing); CGR4 (75 days after sowing)). Lettering (a–f) on bars shows mean separation within columns by one-way Tukey’s HSD at *p* ≤ 0.05.

**Figure 5 molecules-26-02022-f005:**
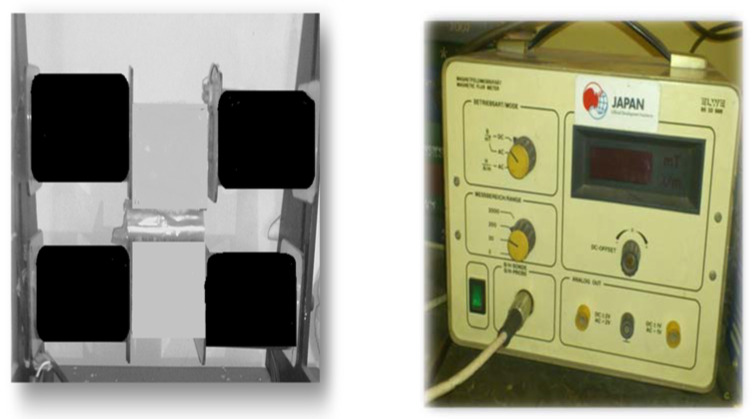
Magnetometer for magnetic seed treatment.

**Table 1 molecules-26-02022-t001:** Effect of magnetic seed stimulation on germination of sunflower Armoni genotype.

Magnetic Field	Time (Min)	MGT (Days)	GI	FG (%)	GE (%)
50 mT	5	9.3 ± 0.75 a	2.19 ± 0.16 ab	70 ± 2.88 b	66 ± 3.46 abc
	10	8.75 ± 0.43 a	2.31 ± 0.11 ab	67 ± 1.73 b	70 ± 2.88 abc
	15	7.96 ± 0.28 ab	2.44 ± 0.06 ab	73 ± 2.3 ab	76 ± 1.73 ab
100 mT	5	6.45 ± 0.25 b	2.41 ± 0.1 ab	76 ± 3.46 ab	73 ± 1.15 ab
	10	5.97 ± 0.62 b	2.69 ± 0.1 a	83 ± 2.3 a	80 ± 4.04 a
	15	7.9 ± 0.45 ab	2.37 ± 0.09 ab	70 ± 2.88 a	65 ± 0.57 bc
150 mT	5	7.85 ± 0.23 ab	2.44 ± 0.07 ab	72 ± 1.15 ab	74 ± 2.88 ab
	10	5.98 ± 0.17 b	2.65 ± 0.11 a	83 ± 1.73 a	78 ± 1.73 ab
	15	6.45 ± 0.19 b	2.05 ± 0.09 b	68 ± 1.15 b	58 ± 4.04 c
Control		9.46 ± 0.26	2.06 ± 0.09	63 ± 1.15	53 ± 1.73
HSD _Interaction_ (MF X T)		4.18	0.50	11.88	14.33

(MGT) Mean germination time; (GI) Germination index; (FG%) Final germination %; (GE) Germination energy. Mean values not sharing the same letters in a column differ significantly at *p* ≤ 0.05. Degree of freedom: Factor 1 (dF1) = 2; Factor 2 (dF2) = 2; dF1*dF2 = 4. HSD = Honestly significant difference.

**Table 2 molecules-26-02022-t002:** Effect of magnetic seed stimulation on seedling growth of sunflower Armoni genotype during germination test.

Magnetic Field	Time (min)	RL (cm)	SL (cm)	RL/SL
50 mT	5	6 ± 0.17 d	7 ± 0.1 d	0.85 ± 0.02 de
	10	8.45 ± 0.19 b	8.13 ± 0.07 bc	1.03 ± 0.01 bc
	15	7.77 ± 0.31 bc	8.74 ± 0.15 ab	0.87 ± 0.02 de
100 mT	5	6.2 ± 0.12 d	7.57 ± 0.15 cd	0.81 ± 0.04 e
	10	10.44 ± 0.42 a	9 ± 0.19 a	1.16 ± 0.02 a
	15	7.5 ± 0.28 bc	7.2 ± 0.11 d	1.04 ± 0.01 bc
150 mT	5	6.8 ± 0.21 cd	7.06 ± 0.2 d	0.95 ± 0.01 cd
	10	9.8 ± 0.17 a	8.9 ± 0.11 a	1.1 ± 0.01 ab
	15	7.8 ± 0.2 bc	7.4 ± 0.17 d	1.06 ± 0.01 ab
Control		5.99 ± 0.08	6.7 ± 0.17	0.89 ± 0.02
HSD _Interaction_ (MF X T)		1.27	0.72	0.10

(RL) Root length; (GI) Shoot length; (RL/SL) Root shoot ratio. Mean values not sharing the same letters in a column differ significantly at *p* ≤ 0.05. Degree of freedom: Factor 1 (dF1) = 2; Factor 2 (dF2) = 2; dF1*dF2 = 4. HSD = Honestly significant difference.

**Table 3 molecules-26-02022-t003:** Effect of magnetic seed stimulation on biochemical attributes of sunflower Armoni genotype during germination test.

Magnetic Field	Time (Min)	α Amylase	TSS	RS
50 mT	5	19.73 ± 0.35 f	21.56 ± 0.76 d	16.43 ± 1.13 e
	10	24.17 ± 0.6 e	27.89 ± 1.72 cd	20.53 ± 0.59 e
	15	33.37 ± 1.31 cd	47.53 ± 1.07 b	38.63 ± 2.8 bc
100 mT	5	22.63 ± 1.5 ef	27.22 ± 2.4 cd	20.65 ± 1.44 e
	10	49.35 ± 2.85 a	57.49 ± 4.24 a	52.14 ± 0.63 a
	15	37.43 ± 0.68 c	46.38 ± 2.47 b	41.64 ± 2.22 b
150 mT	5	26.41 ± 0.17 e	32.1 ± 1.19 c	28.15 ± 1.05 d
	10	43.18 ± 1.23 b	55.46 ± 1.93 ab	47.48 ± 1.44 a
	15	31.79 ± 0.45 d	47.22 ± 0.67 b	34.26 ± 1.84 c
Control		14.04 ± 0.02	31.27 ± 1.27	11.45 ± 0.25
HSD Interaction (MF X T)		4.18	9.74	4.90

α amylase specificity (IU mg^−1^); (TSS) Total soluble sugars (mg g^−1^); (RS) Reducing sugars (mg g^−1^). Mean values not sharing the same letters in a column differ significantly at *p* ≤ 0.05. Degree of freedom: Factor 1 (dF1) = 2; Factor 2 (dF2) = 2; dF1*dF2 = 4. HSD = Honestly significant difference.

**Table 4 molecules-26-02022-t004:** Effect of seed enhancements on yield and yield related attributes of sunflower Armoni genotype under field conditions.

Seed Enhancements	Plant Height (cm)	No. of Achene/Head	Head Diameter (cm)	1000-Achene Weight (g)	Biological Yield (Kg ha^−1^)	Achene Yield (Kg ha^−1^)	Harvest Index (%)
Control	166 ± 0.43 e ^a^	253 ± 1.71 f	6.87 ± 0.12 c	37.59 ± 0.22 d ^a^	6073 ± 63.4 c	1321 ± 20.5 e	21.76 ± 0.37 e
Hydropriming	171 ± 0.45 d	446 ± 1.63 d	7.34 ± 0.91 b	40.12 ± 0.27 c	6306 ± 71.2 c	1432 ± 25.1 d	22.71 ± 0.4 de
MTW (10 min)	176 ± 0.39 b	495 ± 1.41 b	7.52 ± 0.81 b	40.98 ± 0.24 b	6656 ± 63.1 b	1760 ± 19.75 b	26.44 ± 0.28 b
100 mT for 10 min	178 ± 0.32 a	502 ± 1.8 a	7.90 ± 0.07 a	42.57 ± 0.19 a	6977 ± 70.3 a	1973 ± 27.7 a	28.30 ± 0.5 a
150 mT for 10 min	176 ± 0.48 b	487 ± 1.62 c	7.30 ± 0.10 b	35.44 ± 0.29 e	6266 ± 63.5 c	1456 ± 15.5d	23.24 ± 0.31 de
MLE (3%)	173 ± 0.36 c	406 ± 1.59 e	7.39 ± 0.07 b	37.15 ± 0.18 d	6632 ± 67.2 b	1610 ± 25.41 c	24.28 ± 0.23 cd
MLE + MTW	176 ± 0.27 b	495 ± 1.72 b	7.88 ± 0.08 a	40.26 ± 0.23 bc	6728 ± 56.5 ab	1680 ± 20.6 bc	24.98 ± 0.4 bc
HSD at *p* ≥ 0.05	1.82	6.64	0.35	0.82	282.12	93.07	1.66

^a^ Mean separation within columns by one-way Tukey’s HSD at *p* ≤ 0.05. Mean values not sharing the same letters in a column differ significantly at *p* ≤ 0.05. MTW = Magnetically treated water, Min = Minutes, MLE = Moringa leaf extract. HSD = Honestly significant difference.

## Data Availability

Not applicable.

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
