# Peer review of "Magnetic Field Treatments Improves Sunflower Yield by Inducing Physiological and Biochemical Modulations in Seeds"

_molecules, 2021, doi:10.3390/molecules26072022_

Round 1

Reviewer 1 Report

The manuscript is well written. The experimental design is thorough and well laid out.

However, the results as presented are not correct regarding the poorly performed statistical analysis which, of course, can be completed in order to make the paper ready to publish. Please, follow my comments below: Magnetic seed treatment requires a

  1. Models: There should be used 4 different main models with some different evaluations:

Model 1. two-way MANOVA: Magnetic seed treatment study

Two-way: strength of magnetic field with 3 levels + time effect with three levels.

MANOVA (multivariate ANOVA): because the dependent variables (see below) are related variables.

Model 1.1: dependent variables: Germination time, Germination index, Final germination percentage, Germination energy %,

Model 1.2: dependent variables: Root length, Shoot length, Root/Shoot ratio,

Model 1.3: α-amylase specificity, Total soluble sugar, Reducing sugar

Model 2. One-way MANOVA: Priming with Magnetically Treated Water

One-way: treatment with 5 levels

MANOVA (multivariate ANOVA): because the dependent variables (see below) are related variables.

Model 2.1: dependent variables: Emergence time, Emergence energy, Emergence index, Final Emergence

Model 2.2: dependent variables: Root length, Shoot length, Root/Shoot ratio,

Model 2.3: dependent variables: α-amylase activity, Total soluble sugar, Reducing sugar

Model 3. One-way MANOVA random block design: Field study

Block design: because the authors state the design was RCBD.

One-way: treatment with 7 levels

MANOVA (multivariate ANOVA): because the dependent variables (see below) are related variables.

Model 3.1: dependent variables: Emergence time, Emergence index, Final Emergence

Model 3.2: dependent variables: Plant height, Head diameter, # of achene per head, 1000 achene weight, Biological yield, Achene Yield, Harvest Index

Model 4. Two-way ANOVA random block design or One-way Repeated measures ANOVA random block design: Field study

Block design: because the authors state the design was RCBD.

Two-way: treatment with 7 levels + time effect

OR

Block design: because the authors state the design was RCBD.

One-way: treatment with 7 levels

Repeated measure of ANOVA: to measure the within subject time-effect

Model 4.1: dependents variable: Crop growth rate

When significant overall result is detected, between-subjects effects should be analysed with e.g Bonferroni's corrections to avoid familywise error and false significant result.

  1. Assumption check: The above methods require some assumptions (normality, variance homogeneity, or, in case of repeated measures ANOVA, also sphericity) that have to be proved, or, if some assumptions are violated, data management methods should be applied to avoid false significant result detection.
  • Post hoc: At the end, post hoc test is used, but post hoc test method should be chosen according to the statistical method: comparing pairwisely all the 10 levels as if there was an only factor (treatment type) does not provide any information on the individual factor effects. Tukey's post hoc test also requires homogeneity of variances that was not checked. Note that in case homogeneity of variances assumption is violated, instead of Tukey’s, Games-Howell’s post hoc test is recommended because this method can manage the assumption violation problem.
  1. Reporting statistical analysis: When reporting the results, please provide the test value of the statistics, the degrees of freedom (df) together with the p values, because they give information about the factor effects and the sample sizes, moreover, if variance homogeneity is violated, df should be adequately corrected. Test values should be given with two digits. Each factor effect should be reported individually as significant or insignificant together with the interaction effect e.g. in case of two-way models, factor1, factor2, the interaction.

According to the above comments, not only chapter 'Results' needs improvement but also Materials and Methods should be completed with all the methods, assumption check methods and outcomes, data manipulation (if there was any), corrections, post hoc test method etc.

Author Response

Dear Reviewer

            Thank you for your valuable suggestion and comments for the improvement of current manuscript. We have improved the manuscript based your comments. All the mentioned changes have been incorporated in the manuscript. We appreciate for your warm work earnestly and hope that the correction will meet with approval.

On behalf of all co-authors, once again thank you for your valuable efforts.

Best regards,

Ayman El Sabagh

Response to Reviewer

The manuscript is well written. The experimental design is thorough and well laid out.

However, the results as presented are not correct regarding the poorly performed statistical analysis which, of course, can be completed in order to make the paper ready to publish. Please, follow my comments below: Magnetic seed treatment requires a

Comment 1:

  1. Models: There should be used 4 different main models with some different evaluations:

Model 1. two-way MANOVA: Magnetic seed treatment study

Two-way: strength of magnetic field with 3 levels + time effect with three levels.

MANOVA (multivariate ANOVA): because the dependent variables (see below) are related variables.

Model 1.1: dependent variables: Germination time, Germination index, Final germination percentage, Germination energy %,

Model 1.2: dependent variables: Root length, Shoot length, Root/Shoot ratio,

Model 1.3: α-amylase specificity, Total soluble sugar, Reducing sugar

Author’s response:

            Esteemed reviewer, the current study was conducted in two different phases (Lab screening and field study). Lab screening was further divided into two different experiments. For first experiment of lab screening, we have applied two-way multivariate ANOVA (magnetic field treatment x time of exposure) based on your suggestion. Model 1 was executed to the first phase of lab screening. Moreover, we have also edited the statistical analysis section in materials and methods, providing information about each statistical model employed on different experiments. We have also provided the information about statistical model along with each figure and table.

Comment 2:

Model 2. One-way MANOVA: Priming with Magnetically Treated Water

One-way: treatment with 5 levels

MANOVA (multivariate ANOVA): because the dependent variables (see below) are related variables.

Model 2.1: dependent variables: Emergence time, Emergence energy, Emergence index, Final Emergence

Model 2.2: dependent variables: Root length, Shoot length, Root/Shoot ratio,

Model 2.3: dependent variables: α-amylase activity, Total soluble sugar, Reducing sugar

Author’s response:

            Dear reviewer, we have applied One-way MANOVA for second priming experiment of lab screening based on your suggestion. Moreover, we have also edited the statistical analysis section in materials and methods, providing information about each statistical model employed on different experiments. We have also provided the information about statistical model along with each figure and table.

Comment 3:

Model 3. One-way MANOVA random block design: Field study

Block design: because the authors state the design was RCBD.

One-way: treatment with 7 levels

MANOVA (multivariate ANOVA): because the dependent variables (see below) are related variables.

Model 3.1: dependent variables: Emergence time, Emergence index, Final Emergence

Model 3.2: dependent variables: Plant height, Head diameter, # of achene per head, 1000 achene weight, Biological yield, Achene Yield, Harvest Index

Author’s response:

Esteemed reviewer, as mentioned earlier the study was conducted in two phases. First phase (lab study) included bi-factorial ANOVA (magnetic field treatment x time of exposure) and One-way MANOVA (priming treatments). While, the second phase (field study), comprised of best screened treatments from lab, (treatment with 7 levels) was subjected to one-way MANOVA according to your suggestion.

Comment 4:

Model 4. Two-way ANOVA random block design or One-way Repeated measures ANOVA random block design: Field study

Block design: because the authors state the design was RCBD.

Two-way: treatment with 7 levels + time effect

OR

Block design: because the authors state the design was RCBD.

One-way: treatment with 7 levels

Repeated measure of ANOVA: to measure the within subject time-effect

Model 4.1: dependents variable: Crop growth rate

Author’s response: Not applicable.

Comment 5:

When significant overall result is detected, between-subjects effects should be analysed with e.g Bonferroni's corrections to avoid familywise error and false significant result.

  1. Assumption check: The above methods require some assumptions (normality, variance homogeneity, or, in case of repeated measures ANOVA, also sphericity) that have to be proved, or, if some assumptions are violated, data management methods should be applied to avoid false significant result detection.

Author’s response:

            Dear reviewer, to check the normal distribution of data, we conducted Leven’s test of homogeneity of variance using SPSS. The “significance values” obtained were higher than 0.05, indicating non-significance of the model. Which means, the variances were homogenous, depicting normal distribution of data.

Comment 6:

  • Post hoc: At the end, post hoc test is used, but post hoc test method should be chosen according to the statistical method: comparing pairwisely all the 10 levels as if there was an only factor (treatment type) does not provide any information on the individual factor effects. Tukey's post hoc test also requires homogeneity of variances that was not checked. Note that in case homogeneity of variances assumption is violated, instead of Tukey’s, Games-Howell’s post hoc test is recommended because this method can manage the assumption violation problem.
  1. Reporting statistical analysis: When reporting the results, please provide the test value of the statistics, the degrees of freedom (df) together with the p values, because they give information about the factor effects and the sample sizes, moreover, if variance homogeneity is violated, df should be adequately corrected. Test values should be given with two digits. Each factor effect should be reported individually as significant or insignificant together with the interaction effect e.g. in case of two-way models, factor1, factor2, the interaction.

According to the above comments, not only chapter 'Results' needs improvement but also Materials and Methods should be completed with all the methods, assumption check methods and outcomes, data manipulation (if there was any), corrections, post hoc test method etc.

Author’s response:

In our experiments, we normally used Tukey’s test with the consult of our statistician. In order to maintain the length of manuscript, we did not provide too much information regarding test values of statistics and degree of freedom however we have provided the p values in the last row of table 1. We can provide additional details regarding statistical analysis in separate file and submit as repository file.  

Reviewer 2 Report

The paper entitled "Magnetic Field Treatments Improve Sunflower Yield by Inducting Physiological and Biochemical Modulations in Seeds” is focused on stimulation of oilseeds and comparing various priming methods and their influence on germination and growth parameters. The abstract’s length is suitable, language is proper.

Topic is interesting and presented results show an improvement in comparison to the control.

 However I have several questions, which should be clarified:

  1. How many seeds were in each lot, how many plants were observed after germination?
  2. What are the properties of magnetically treated water? Did you perform some analysis concerning composition, radicals, etc.? What is the reason of its action?
  3. Please describe hydropriming procedure
  4. Please improve Fig.6 (control in the middle graph)

Author Response

Dear Reviewer

            Thank you for your valuable suggestion and comments for the improvement of current manuscript. We have improved the manuscript based your comments. All the mentioned changes have been incorporated in the manuscript. We appreciate for your warm work earnestly and hope that the correction will meet with approval.

On behalf of all co-authors, once again thank you for your valuable efforts.

Best regards,

Ayman El Sabagh

Response to Reviewer

The paper entitled "Magnetic Field Treatments Improve Sunflower Yield by Inducting Physiological and Biochemical Modulations in Seeds” is focused on stimulation of oilseeds and comparing various priming methods and their influence on germination and growth parameters. The abstract’s length is suitable, language is proper.

Topic is interesting and presented results show an improvement in comparison to the control.

 However I have several questions, which should be clarified:

Comment 1: How many seeds were in each lot, how many plants were observed after germination?

Authors reply: Dear reviewer, thank you for your appreciated comments and suggestions. We have added the information about seed weight used for each treatments. We also incorporated the number of seeds sown. Moreover, we have also included the sample size for determination of growth, morphological, yield and biochemical attributes.

Comment 2: What are the properties of magnetically treated water? Did you perform some analysis concerning composition, radicals, etc.? What is the reason of its action?

Authors reply: The properties of magnetically treated water are well reputed in the literature. We have only tested for pH, conductivity and dissolved oxygen concentration of magnetically treated water. An increase of pH upto 0.06, conductivity upto 0.05 uS/cm, and dissolved oxygen concentration upto 8.0% were recorded.

Comment 3: Please describe hydropriming procedure

Authors reply: We have explained the procedure for hydropriming according to your suggestion.

Comment 4: Please improve Fig.6 (control in the middle graph)

Authors reply: We have edited the figure according to your suggestion.

Reviewer 3 Report

The manuscript describes the use of the static magnetic field (MF) exposure, magnetically treated water and moringa leaf extract for improve germination, length root and shoot, α-amylase specificity, quantity of sugars of the sunflower seeds. Despite the interesting results and well-chosen statistical criteria (ANOVA, Tukey’s HSD), the article needs revision.

The main declared result is devoted to the effect of a MF. The main declared result is devoted to the effect of a magnetic field. However the magnetic field (MF) exposure system is poorly described in the article. The authors refer on past paper [14], but that article has little information about MF exposure too. It would be nice to give photo or diagram exposure system. There is no information on how large the MT gradients are at the exposure site and about low frequency MF at the site of exposure. There is no description in what magnetic conditions the control was exposure.

It would be nice to give information (accuracy, sensitivity, range, linear dimensions) about the magnetic sensor ELWE 8533996 that was used for the MF measurement, since there is little information about it on the Internet.

It’s unclear why the MF inductions 50 mT, 100 mT, 150mT (in the previous publication 60 mT, 120 mT, 180mT [14]) were chosen?

It’s unclear why the MF inductions 210 mT for water exposure were chosen?

Please add and clarify two points in the discussion. At first why a 15-minute MF exposure with an induction of 100 mT practically does not cause affects as compared with 5 and 10 minutes exposures? Secondly, why does water have a biotrophic effect for several hours after 10 minutes of magnetic treatment? It would be nice if you could suggest a physical mechanism for this phenomenon. By the way, the materials and methods do not contain information on the purity of the distilled water used.

Could you give definitions in the form of formulas for all the measured parameters used in the article: Mean germination time, Mean emergence time, etc. This should be done to compare the "Mean germination time" and "Mean emergence time" for the Control in Figures 1 and 4.

P12. Line 262:

“Cremilingen”

P9. Line 134 Figure 7:

“CGR1, 30 days after swing”

P9. Line 134 Figure 7

Explain what the labels “MTS 100” and “MTS 150” in the figure.

P10. Line 146 Table 1

Explain what the means “LSD” in the table.

Author Response

Dear Reviewer

            Thank you for your valuable suggestion and comments for the improvement of current manuscript. We have improved the manuscript based your comments. All the mentioned changes have been incorporated in the manuscript. We appreciate for your warm work earnestly and hope that the correction will meet with approval.

On behalf of all co-authors, once again thank you for your valuable efforts.

Best regards,

Ayman El Sabagh

Response to Reviewer

Comment 1: The manuscript describes the use of the static magnetic field (MF) exposure, magnetically treated water and moringa leaf extract for improve germination, length root and shoot, α-amylase specificity, quantity of sugars of the sunflower seeds. Despite the interesting results and well-chosen statistical criteria (ANOVA, Tukey’s HSD), the article needs revision.

Author’s response: Dear reviewer, thank you for your appreciated comments and suggestion.

Comment 2: The main declared result is devoted to the effect of a MF. The main declared result is devoted to the effect of a magnetic field. However the magnetic field (MF) exposure system is poorly described in the article. The authors refer on past paper [14], but that article has little information about MF exposure too. It would be nice to give photo or diagram exposure system. There is no information on how large the MT gradients are at the exposure site and about low frequency MF at the site of exposure. There is no description in what magnetic conditions the control was exposure.

Author’s response: The photo of diagram exposure system is added.

Magnetometer for magnetic seed treatment

Comment 3: It would be nice to give information (accuracy, sensitivity, range, linear dimensions) about the magnetic sensor ELWE 8533996 that was used for the MF measurement, since there is little information about it on the Internet.

Author’s response: In full wave rectification four diodes in a bridge configuration were used with step down transformer, which convert any AC source into pulsating DC. Four rectangular coils which are not Helmholtz coils were constructed using enameled copper wire of thickness 0.42 mm, each coil having 4000 turns. These coils are wound on two soft iron bars (electromagnet, having dimension 40 cm x 3.5 cm) kept one above the other (two coils wound on each bar), their ends held by metallic supports. These coils were connected in series in order to maintain the current same in all coils and fed through a power source using a variable transformer (0 V-260 V AC). A 50 Hz full wave rectified sinusoidal voltage was fed to the coils. When electric current is passed through these coils, a non-uniform magnetic field is produced between the air gaps of two iron bars, the magnetic field (MF) is varied by changing the current through the coils. Moreover, on each side of magnetic field setup, 13 magnets were attached. A pvc pipe was passed through this magnets and the measured magnetic field between these magnet was 210 mT. When water was passed through this magnetic field, as water contain sodium chloride, it converted into ionic layers of sodium on the negative side of the magnet and chlorine on the positive side of the magnets also molecular cluster of H2O becomes smaller as the water passed through these magnetic field. 

Comment 4: It’s unclear why the MF inductions 50 mT, 100 mT, 150mT (in the previous publication 60 mT, 120 mT, 180mT [14]) were chosen?

Author’s response: Dear reviewer, we want to check the positive/negative effects of low and high MF inductions on sunflower. Therefore, we included both high and low MF inductions for experimentation. Moreover, we have tested all the selected systematic MF treatments in lab as well as in field (selected MF treatments based on lab results) to determine the best MF treatment for sunflower. Previous study evaluated the effect of MF treatments on pea.

Comment 5: It’s unclear why the MF inductions 210 mT for water exposure were chosen?

Author’s response: Esteemed reviewer, on each side of magnetic field setup, 13 magnets were attached. A pvc pipe was passed through this magnets and the measured magnetic field between these magnet was 210 mT. When water was passed through this magnetic field, as water contain sodium chloride, it converted into ionic layers of sodium on the negative side of the magnet and chlorine on the positive side of the magnets also molecular cluster of H2O becomes smaller as the water passed through these magnetic field. 

Comment 6: Please add and clarify two points in the discussion. At first why a 15-minute MF exposure with an induction of 100 mT practically does not cause affects as compared with 5 and 10 minutes exposures? Secondly, why does water have a biotrophic effect for several hours after 10 minutes of magnetic treatment? It would be nice if you could suggest a physical mechanism for this phenomenon. By the way, the materials and methods do not contain information on the purity of the distilled water used.

 Author’s response: Seeds treated with 100 mT for 15 min did not perform well as compared to 10 min exposure time as indicated by lower value of amylase activity in seeds exposed to 15 min as this enzyme is responsible for conversion of starch into simple sugars and an indication of better germination capacity. Similarly, seeds treated with 100 mT for 15 min also showed low germination and emergence potential and took more time to germinate as compared to seeds exposed to 10 min exposure time. Moreover, during this study, germination assays were conducted on seeds after respective magnetic treatments without any delay. Otherwise the effect of treatment was not evaluated. However, it is important to investigate the performance of magnetically treated seed at different time intervals after storage in the ambient conditions. 

Comment 7: Could you give definitions in the form of formulas for all the measured parameters used in the article: Mean germination time, Mean emergence time, etc. This should be done to compare the "Mean germination time" and "Mean emergence time" for the Control in Figures 1 and 4.

Author’s response: We have provided the formulas for all the measured parameters used in the article.

Comment 8: P12. Line 262: “Cremilingen”

Author’s response: We have corrected the spelling mistake. “Cremilingen” has been changed to “Cremlingen”

Comment 9: P9. Line 134 Figure 7: “CGR1, 30 days after swing”

Author’s response: We have corrected according to your suggestion.

Comment 10: P9. Line 134 Figure 7 Explain what the labels “MTS 100” and “MTS 150” in the figure.

Author’s response: Sorry for inconvenience. We have corrected the typographic mistakes in the legends of figure 7.

Comment 11: P10. Line 146 Table 1 Explain what the means “LSD” in the table.

Author’s response: Dear reviewer, we have explained “LSD” according to your suggestion.

Round 2

Reviewer 1 Report

I appreciate the effort of the authors for improving the statistical evaluation of their experiment, however, I can accept neither the corrections nor their response.

I ask the authors to provide the assumtion check information not only in their response letter but also in 'Statistical methods'.

In case a MANOVA tests were run, the authors need to mention the overall tests results (e.g. Wilk's lambdas with p) and if the test were significant, the follow-up analysis with e.g. Bonferroni"s Type 1 error correction. The space-sparing reason is also unacceptable if it is about reporting statistical results. As I required in my previous review, test values of the statistics (F), the degrees of freedom (df1,df2) together with the p values have to be given, because they provide information about the factor effects and the sample sizes. Each factor effect should be reported individually as significant or insignificant together with the interaction effect e.g. in case of two-way models, factor1, factor2, the interaction. (They do not require too much space.)

I also asked the authors to run a detailled post hoc test from which one can see the effects of the two factors separately.

"Comparing pairwisely all the 10 levels as if there was an only factor (treatment type) does not provide any information on the individual factor effects." 

I provide here a reference from mdpi publisher that shows how to report a two-way analysis post hoc test result:

https://doi.org/10.3390/agriculture10040094

Author Response

Esteemed Reviewer

            Once again, thank you for your valuable suggestion and comments for the improvement of current manuscript. We have improved the manuscript based your comments. All the mentioned changes have been incorporated in the manuscript. We appreciate for your warm work earnestly and hope that the correction will meet with approval.

On behalf of all co-authors, thank you for your valuable efforts.

Best regards,

Ayman El Sabagh

Response to Reviewer

Comment 1: I ask the authors to provide the assumption check information not only in their response letter but also in 'Statistical methods'.

Author’s response: Esteemed reviewer, we have provided the assumption check information in statistical methods section accordingly.

Comment 1: In case a MANOVA tests were run, the authors need to mention the overall tests results (e.g. Wilk's lambdas with p) and if the test were significant, the follow-up analysis with e.g. Bonferroni"s Type 1 error correction. The space-sparing reason is also unacceptable if it is about reporting statistical results. As I required in my previous review, test values of the statistics (F), the degrees of freedom (df1, df2) together with the p values have to be given, because they provide information about the factor effects and the sample sizes. Each factor effect should be reported individually as significant or insignificant together with the interaction effect e.g. in case of two-way models, factor1, factor2, the interaction. (They do not require too much space.). I also asked the authors to run a detailed post hoc test from which one can see the effects of the two factors separately.

"Comparing pair wisely all the 10 levels as if there was an only factor (treatment type) does not provide any information on the individual factor effects."

I provide here a reference from mdpi publisher that shows how to report a two-way analysis post hoc test result:

https://doi.org/10.3390/agriculture10040094

Author’s response: For first part of lab screening we have followed your suggestion and the link (https://doi.org/10.3390/agriculture10040094) provided by you. We have modified our results section based on your suggestion.

Table 1: Effect of magnetic seed stimulation on germination of sunflower hybrid Armoni.

Magnetic Field

Time (min)

MGT (days)

GI

FG (%)

GE (%)

50 mT

5

9.3 ± 0.75 a

2.19 ± 0.16 ab

70 ± 2.88 b

66 ± 3.46 abc

10

8.75 ± 0.43 a

2.31 ± 0.11 ab

67 ± 1.73 b

70 ± 2.88 abc

15

7.96 ± 0.28 ab

2.44 ± 0.06 ab

73 ± 2.3 ab

76 ± 1.73 ab

100 mT

5

6.45 ± 0.25 b

2.41 ± 0.1 ab

76 ± 3.46 ab

73 ± 1.15 ab

10

5.97 ± 0.62 b

2.69 ± 0.1 a

83 ± 2.3 a

80 ± 4.04 a

15

7.9 ± 0.45 ab

2.37 ± 0.09 ab

70 ± 2.88 a

65 ± 0.57 bc

150 mT

5

7.85 ± 0.23 ab

2.44 ± 0.07 ab

72 ± 1.15 ab

74 ± 2.88 ab

10

5.98 ± 0.17 b

2.65 ± 0.11 a

83 ± 1.73 a

78 ± 1.73 ab

15

6.45 ± 0.19 b

2.05 ± 0.09 b

68 ± 1.15 b

58 ± 4.04 c

Control

9.46 ± 0.26

2.06 ± 0.09

63 ± 1.15

53 ± 1.73

HSD Interaction (MF X T)

 4.18

 0.50

 11.88

 14.33

(MGT) Mean germination time; (GI) Germination index; (FG %) Final germination %; (GE) Germination energy.

Mean values not sharing the same letters in a column differ significantly at p ≤ 0.05.

Degree of freedom: Factor 1 (dF1) = 2; Factor 2 (dF2) = 2; dF1*dF2= 4.

HSD = honestly significant difference.

Table 2: Effect of magnetic seed stimulation on seedling growth of sunflower hybrid Armoni during germination test.

Magnetic Field

Time (min)

RL (cm)

SL (cm)

RL/SL

50 mT

5

6 ± 0.17 d

7 ± 0.1 d

0.85 ± 0.02 de

10

8.45 ± 0.19 b

8.13 ± 0.07 bc

1.03 ± 0.01 bc

15

7.77 ± 0.31 bc

8.74 ± 0.15 ab

0.87 ± 0.02 de

100 mT

5

6.2 ± 0.12 d

7.57 ± 0.15 cd

0.81 ± 0.04 e

10

10.44 ± 0.42 a

9 ± 0.19 a

1.16 ± 0.02 a

15

7.5 ± 0.28 bc

7.2 ± 0.11 d

1.04 ± 0.01 bc

150 mT

5

6.8 ± 0.21 cd

7.06 ± 0.2 d

0.95 ± 0.01 cd

10

9.8 ± 0.17 a

8.9 ± 0.11 a

1.1 ± 0.01 ab

15

7.8 ± 0.2 bc

7.4 ± 0.17 d

1.06 ± 0.01 ab

Control

5.99 ± 0.08

6.7 ± 0.17

0.89 ± 0.02

HSD Interaction (MF X T)

 1.27

 0.72

 0.10

(RL) Root length; (GI) Shoot length; (RL/SL) Root shoot ratio.

Mean values not sharing the same letters in a column differ significantly at p ≤ 0.05.

Degree of freedom: Factor 1 (F1) = 2; Factor 2 (F2) = 2; dF1*dF2= 4.

HSD = honestly significant difference.

Table 3: Effect of magnetic seed stimulation on biochemical attributes of sunflower hybrid Armoni during germination test.

Magnetic Field

Time (min)

α amylase

TSS

RS

50 mT

5

19.73 ± 0.35 f

21.56 ± 0.76 d

16.43 ± 1.13 e

10

24.17 ± 0.6 e

27.89 ± 1.72 cd

20.53 ± 0.59 e

15

33.37 ± 1.31 cd

47.53 ± 1.07 b

38.63 ± 2.8 bc

100 mT

5

22.63 ± 1.5 ef

27.22 ± 2.4 cd

20.65 ± 1.44 e

10

49.35 ± 2.85 a

57.49 ± 4.24 a

52.14 ± 0.63 a

15

37.43 ± 0.68 c

46.38 ± 2.47 b

41.64 ± 2.22 b

150 mT

5

26.41 ± 0.17 e

32.1 ± 1.19 c

28.15 ± 1.05 d

10

43.18 ± 1.23 b

55.46 ± 1.93 ab

47.48 ± 1.44 a

15

31.79 ± 0.45 d

47.22 ± 0.67 b

34.26 ± 1.84 c

Control

14.04 ± 0.02

31.27 ± 1.27

11.45 ± 0.25

HSD Interaction (MF X T)

 4.18

 9.74

 4.90

α amylase specificity (IU mg-1); (TSS) Total soluble sugars (mg g-1); (RS) Reducing sugars (mg g-1).

Mean values not sharing the same letters in a column differ significantly at p ≤ 0.05.

Degree of freedom: Factor 1 (F1) = 2; Factor 2 (F2) = 2; dF1*dF2= 4.

HSD = honestly significant difference.

We have replaced the figures with tables following the manuscript link provided by you. We have modified the statistical analysis section accordingly. 

Reviewer 2 Report

I recommend to accept this paper.

Author Response

Dear Reviewer

            Thank you for your valuable suggestion and comments for the improvement of current manuscript.

On behalf of all co-authors, once again thank you for your valuable efforts.

Best regards,

Ayman El Sabagh

Reviewer 3 Report

I advise the authors to include in the text of the article the information presented in the answers to my questions (questions regarding different inductances, questions about the sensor, the question about the purity of water). If I received answers to these questions, then why did the rest of the readers not receive this information? 

Author Response

Dear Reviewer

            Thank you for your valuable suggestion and comments for the improvement of current manuscript. We have improved the manuscript based your comments. All the mentioned changes have been incorporated in the manuscript. We appreciate for your warm work earnestly and hope that the correction will meet with approval.

On behalf of all co-authors, once again thank you for your valuable efforts.

Best regards,

Ayman El Sabagh

Response to Reviewer

Comment: I advise the authors to include in the text of the article the information presented in the answers to my questions (questions regarding different inductances, questions about the sensor, the question about the purity of water). If I received answers to these questions, then why did the rest of the readers not receive this information?

Author’s response: Esteemed reviewer, sorry for inconvenience. We have incorporated all the information in the manuscript.